# Representational Curvature Modulates Behavioral Uncertainty in Large Language Models

**Jack G. King** [1]  **Evelina Fedorenko** [1]  **Eghbal A. Hosseini** [1] [2]

## Abstract

Temporal straightening describes how, across layers in autoregressive LLMs, the trajectory traced by the token representations of an input sequence becomes straighter, potentially enabling next-token prediction by linear extrapolation. However, a direct link between this trajectory and token-level behavior has been missing. We provide such a link by relating *contextual curvature*—a geometric measure of how sharply the representational trajectory bends over recent context—to next-token entropy. Across two models (GPT-2 XL and Pythia-2.8B), contextual curvature is correlated with entropy, and this relationship emerges during training. Perturbation experiments reveal selective dependence: manipulating curvature through trajectory-aligned interventions reliably modulates entropy, while geometrically misaligned perturbations have no effect. Finally, regularizing representations to be straighter during training modestly reduces token-level entropy without degrading validation loss. These results identify trajectory curvature as a task-aligned representational feature that influences behavioral uncertainty in LLMs.

## 1. Introduction

Large language models (LLMs) are general-purpose neural networks that autoregressively integrate their inputs into a predictive distribution over the next input (Radford et al., 2017). Interpretability research has approached LLMs from several angles: probing what information their representations encode (Liu et al., 2019; Hewitt & Manning, 2019),

dissecting the circuits and features that implement specific computations (Olsson et al., 2022; Templeton et al., 2024), and developing normative accounts of what those computations should look like (Simon et al., 2024; Jin et al., 2025). While most of these studies work to characterize the representations and computations that enable model behavior, fewer have asked what the next-token prediction objective itself demands of those representations. Grounding representational hypotheses in the mechanics of this objective is critical, since it is the pressure that shapes them in the first place.

One candidate framework for linking objective to representation in temporal prediction comes from computational neuroscience. Directly predicting raw sensory input is intractable due to highly nonlinear relationships between successive states (e.g., pixel values across video frames), but transforming sequences into appropriate neural representations can organize temporal trajectories into geometrically predictable patterns. In particular, Hénaff et al. (2019) demonstrated that perceptual representations of visual sequences are straightened in the human visual system, making it easier to extrapolate to future states. Evidence for this temporal straightening principle extends to primate visual cortex (Hénaff et al., 2021), and there is recent evidence that predictive learning directly reshapes representational geometry in the brain (Greco et al., 2024).

Whereas the visual system's predictive role must be inferred, LLMs are trained explicitly to predict upcoming tokens, making them a direct test of what representations emerge under a prediction objective. Language also carries nonlinear temporal structure at multiple scales (Gibson, 2000; Levy, 2008), making it a domain where straightening would be especially valuable. Indeed, Hosseini & Fedorenko (2023) showed that in autoregressive language models, trajectory curvature is near chance in early layers and decreases to a minimum in middle layers, particularly for more predictable sequences. Whether this geometry actually shapes token-level predictions, however, remains an open question.

We address this question by relating *contextual curvature*—a measure of how sharply the representational trajectory bends over recent context—to next-token entropy, a measure of model uncertainty (Geng et al., 2024). The temporal

[1]Department of Brain and Cognitive Sciences, Massachusetts Institute of Technology, Cambridge, MA, USA [2]Work performed at MIT; currently at Google DeepMind. Correspondence to: Jack King <jackking@mit.edu>, Eghbal A. Hosseini <ehoseini@mit.edu>.

*Proceedings of the $43^{rd}$ International Conference on Machine Learning*, Seoul, South Korea. PMLR 306, 2026. Copyright 2026 by the author(s).

straightening hypothesis implies that when the representational trajectory is straight, future states are predictable from past states; when it bends, the next state is harder to predict (temporal straightening schematic in Fig. 1). We provide three lines of evidence for this relationship. First, we show that curvature is predictive of next-token entropy, and that this relationship emerges over the course of training. Second, we demonstrate that targeted perturbations along the representation trajectory modulate entropy more effectively than nonspecific perturbations. Finally, we show that auxiliary objectives targeting curvature can reshape the entropy of the output distribution without degrading task performance. Together, these results support trajectory curvature as a task-aligned representational feature influencing output uncertainty, and provide further evidence that temporal straightening could be a functional solution to next-token prediction.

## 2. Methods

### 2.1. Models

We analyzed two open-source autoregressive transformer language models: GPT-2 XL (Radford et al., 2019) and Pythia-2.8B (Biderman et al., 2023). While both models were trained with a next-token prediction objective, they differ in architecture, parameter count, and pretraining data (Table 1). We selected them to span distinct positional encoding schemes (learned vs. rotary), training corpora, and model scale. In addition, Pythia's open-source training checkpoints allow us to track representational changes over the course of training, and we use the GPT-2 Small architecture for our own controlled training experiments.

### 2.2. Datasets

**long-context:** LAMBADA (Paperno et al., 2016) consists of narrative passages selected so that humans can predict the final word only when given the full passage, not when shown only the immediately preceding sentence. We use this dataset because it emphasizes long-range contextual dependencies, providing a natural setting in which the structure of the contextual representation leading up to a token is likely to play a critical role in next-token prediction. This makes it well-suited for evaluating how geometric properties of internal representations relate to uncertainty and prediction. We use the term long-context to refer to this dataset.

**short-context:** We selected sentences from the Universal Dependencies corpus, which spans diverse topics such as books, newspapers, and web-based sources. Sentences were filtered to include only those with the 100K most frequent English nouns, removing abbreviations. To identify sentences with balanced linguistic dependencies, we limited

*Table 1.* Architectural properties of analyzed models.

| Model | Dim | Layers | Heads | Params | Pos. Embd. |
|---|---|---|---|---|---|
| GPT-2 XL | 1600 | 48 | 25 | 1.5B | Learned |
| Pythia-2.8B | 2560 | 32 | 32 | 2.8B | Rotary |

sentence length to 10–30 tokens. This process yielded 5,815 sentences, which we refer to as the short-context dataset.

### 2.3. Contextual Curvature

Following Hosseini & Fedorenko (2023), we computed curvature based on neural trajectories of token activations within a sequence. In our formulation, each token is treated as a point in high-dimensional representation space, and curvature quantifies how sharply the trajectory bends at that point.

Given a sequence of tokens $w_1, w_2, ..., w_n$, we extracted their hidden states from the residual stream following each transformer layer, beginning with the first contextualized layer $L_0$. Denote the activation at position $k$ in layer $L_p$ as $x_k^p$. We computed first-order difference vectors as:

$$v_k^p = x_{k+1}^p - x_k^p$$

Curvature at position $k$ was then defined as the angle between adjacent difference vectors (Hosseini & Fedorenko, 2023):

$$c_k^p = \arccos \left( \frac{v_{k+1}^p \cdot v_k^p}{\|v_{k+1}^p\| \, \|v_k^p\|} \right)$$

We define the contextual curvature associated with token $w_k$ at layer $p$ as the average curvature over a backward-looking window. We chose a window of size three because window sizes greater than three did not add predictive power to our regression modeling (Fig. A5):

$$C_k^p = \frac{1}{3} \sum_{i=k-4}^{k-2} c_i^p$$

This provides a localized, contextual measure of changes in direction in the representational trajectory leading up to a token.

### 2.4. Next-Token Entropy

To quantify model uncertainty at each token position, we computed next-token entropy from the model's output logits (Geng et al., 2024). For a given position $n$, *next-token entropy* is defined as:

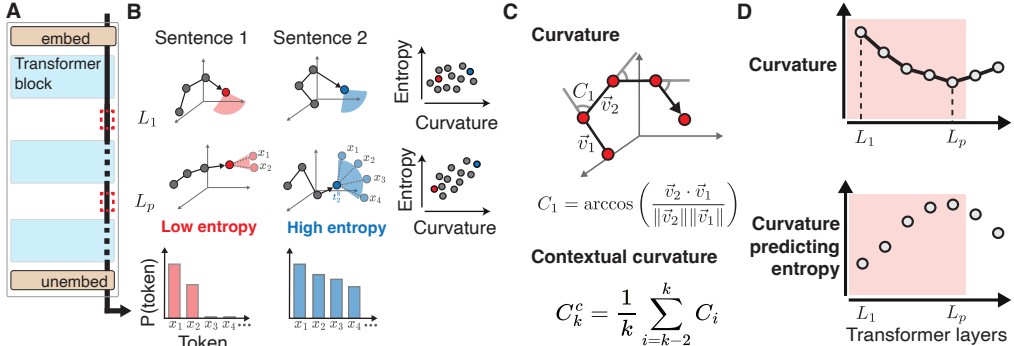

*Figure 1.* **(A)** Schematic of the LLM architecture, focusing on the token representation in the residual stream after each transformer block. **(B)** Hypothesized relationship between the input, internal representation, and output for an easy-to-predict sentence (Sentence 1) versus a hard-to-predict sentence (Sentence 2). A predictable sentence is expected to have a straighter internal representation (lower curvature). Consequently, the set of possible next tokens is smaller, which leads to lower entropy. Conversely, a hard-to-predict sentence is expected to show higher curvature and result in higher entropy. **(C)** The formulation of contextual curvature used to predict token entropy. **(D)** The hypothesized relationship between the change in average curvature across layers and the corresponding change in how well contextual curvature predicts token entropy.

$$H(w_n) = -\sum_i P(w_i \mid w_{1:n-1}) \log_2 P(w_i \mid w_{1:n-1})$$

where $P(w_i \mid w_{1:n-1})$ is the probability assigned to token $w_i$ given the preceding context. These probabilities were obtained by applying a softmax over the model's vocabulary logits at position $n$.

Entropy was computed for each token at positions $n > 6$ to ensure sufficient context. Higher entropy indicates greater uncertainty in next-token prediction, while lower entropy reflects more confident, peaked distributions.

### 2.5. Analysis of Correlation Between Curvature and Entropy

To quantify the linear relationship between internal representation and predictive uncertainty, we use 10-fold cross-validated ordinary least squares regression to predict next-token entropy from scalar representational features.

Our primary predictor is contextual curvature. For each token, we compute its contextual curvature and the entropy of the subsequent token distribution. In each fold, we fit an OLS model with an intercept on the training split and generate predictions on the held-out split. Predictive performance is quantified by the Pearson correlation ($r$) between predicted and observed entropy within each test fold.

Although this setting involves a single scalar predictor, we adopt a cross-validated protocol to estimate the association out of sample and to ensure comparability with analyses involving additional predictors.

To summarize performance, we compute the Pearson correlation separately for each fold, apply a Fisher $z$-transform,

average in $z$-space, and transform back to obtain a pooled estimate of the expected out-of-sample correlation. We estimate uncertainty using a $t$-based 95% confidence interval over the fold-wise Fisher-transformed correlations (df= 9), and report these intervals as error bars in figures.

The same procedure is repeated using contextual magnitude and contextual distance as predictors, reported in the Appendix (Fig. A1). Additional analyses controlling for unigram probability are provided in the Appendix as well (Fig. A2).

### 2.6. Perturbation Experiments

To evaluate the role of contextual curvature in token-level behavior, we designed perturbation experiments in which we applied localized modifications to the residual stream at individual token positions, inducing controlled changes in contextual curvature ($\Delta C$). We then measured the resulting changes in next-token entropy ($\Delta H$).

For each token position $k$, we extracted the residual-stream activation $x_k^p \in \mathbb{R}^d$ from an intermediate layer. Perturbations were applied additively,

$$\tilde{x}_k^p = x_k^p + \delta,$$

and the forward pass was continued from $\tilde{x}_k^p$ while keeping all other activations fixed. All perturbation experiments were conducted with the long-context dataset at intermediate layers, where we suspect extrapolation might occur (Fig. 2: 21 for GPT-2 XL, 11 for Pythia-2.8B). Perturbations are scaled relative to local trajectory geometry: $|\delta| = 0.2|v_k^p|$, where $v_k^p = x_{k+1}^p - x_k^p$ is the displacement to the next token. This ties the perturbation scale to the intrinsic step size of the trajectory at each position, ensuring comparable relative effects across tokens and layers.

We considered five perturbation types, differing only in the geometric subspace from which $\delta$ was drawn. These conditions form a ladder of increasing alignment with the model's representational trajectory: full-space and random-subspace are trajectory-agnostic; activation-subspace controls for low-rank, data-aligned structure without trajectory alignment; trajectory- and planar-subspaces impose trajectory alignment at increasing specificity. Activation-subspace rules out the alternative that any low-dimensional, data-relevant direction would suffice. For the subspace-based perturbations, we fix the subspace dimensionality to $m = 2$ for parity with the planar subspace; results are robust for $m \in \{2, 5, 10\}$.

1. **Full-space:** directions drawn from the unit sphere in $\mathbb{R}^d$.

2. **Random-subspace:** directions drawn from a randomly oriented $m$-dimensional subspace of $\mathbb{R}^d$. Controls for dimensionality while remaining trajectory-agnostic.

3. **Activation-subspace:** directions drawn from the $m$-dimensional subspace defined by the top principal components of residual-stream activations at layer $p$. Captures the dominant directions of variation in individual representations, but is not aligned with trajectories.

4. **Trajectory-subspace:** directions drawn from a per-sample $m$-dimensional subspace defined by the top principal components of that sample's own difference vectors $x_{i+1}^p - x_i^p$ for $i = 0, \ldots, k$.

5. **Planar-subspace:** directions restricted to the two-dimensional subspace spanned by the two most recent difference vectors for each token. This is the tightest possible alignment with the plane in which contextual curvature is defined; it is a special case of trajectory-subspace using only the last two difference vectors rather than PCs over the full recent path.

For each perturbation, we recomputed contextual curvature $\tilde{C}_k^p$ and next-token entropy $\tilde{H}(w_{k+1})$, and defined

$$\Delta C = \tilde{C}_k^p - C_k^p, \qquad \Delta H = \tilde{H}(w_{k+1}) - H(w_{k+1}).$$

For each token, we sampled $N = 300$ perturbations and computed the Pearson correlation between $\Delta C$ and $\Delta H$ across perturbations. These correlations were averaged across tokens to obtain a mean effect size for each perturbation family, with 95% confidence intervals computed by bootstrap resampling over tokens.

To isolate the effect of geometric alignment from perturbation scale, we applied importance reweighting to match the marginal distribution of $|\Delta C|$ across perturbation families before computing correlations (see Appendix A.4).

## 2.7. Training with Curvature Regularization

To examine how training shapes representational geometry, we trained a 12-layer GPT-2 model (GPT-2 Small) with a 512-token context window from scratch. The training dataset consisted of BookCorpus and English Wikipedia, combined in a 1:3 ratio, totaling 100 million tokens. Models were trained on the next-token prediction objective with randomly initialized weights, a batch size of 128, and a fixed validation set used across all runs. Training was terminated at the point of best validation loss.

To directly influence internal geometry, we introduced an auxiliary loss that penalized mean trajectory curvature. Specifically, we added a regularization term at layers 7 and 8 of the form:

$$\mathcal{L}_{\text{curv}}^p = \frac{1}{n} \lambda \sum_{k=0}^{n} c_k^p$$

where $c_k^p$ is the angle-based curvature at position $k$ in layer $p$, $n$ is the number of curves, and $\lambda$ is a weight linearly annealed from 0 to its maximum value ($1 \times 10^{-2}$) over the course of training. The total loss combined the next-token cross-entropy loss with this curvature penalty. Including the curvature penalty resulted in an *untangled* model, whereas including the negative of the curvature penalty resulted in a *tangled* model. Layers 7 and 8 were chosen because these are the layers where trajectories are typically straightest and where curvature–entropy coupling is strongest, suggesting these layers are where the structure of the trajectory most impacts behavior. Furthermore, we found that training with regularization at other layers consistently increased validation loss, suggesting middle layers are also where representational geometry is most amenable to being reshaped without disrupting the model's predictive capacity.

Optimization was performed using AdamW with a learning rate of $3 \times 10^{-4}$. Gradient clipping with a max norm of 1.0 was used to promote stability.

## 2.8. Evaluation of Regularized Models

For each dataset, we first identified the subset of sentences that exhibited both (1) increased curvature in the tangled models relative to baseline, and (2) decreased curvature in the untangled models relative to baseline. This ensured that the auxiliary loss had a consistent directional effect on those sentences.

For each sentence in this subset, we computed the token-wise difference in next-token entropy between each auxiliary-loss model and its corresponding baseline. This was done independently for three models per condition (baseline, untangled, tangled), each trained with a different random seed.

# 3. Results

### 3.1. Contextual Curvature Predicts Token Entropy

We start by establishing a link between contextual curvature—a representational measure—and entropy—a behavioral readout of uncertainty. We quantify this relationship using cross-validated linear regression between token-wise contextual curvature and the entropy of the subsequent predictive distribution. Contextual curvature in the middle layers was positively correlated with entropy. To assess generality, we evaluate this effect across datasets and across two autoregressive models—GPT-2 XL (Radford et al., 2019) and Pythia (Biderman et al., 2023)—which differ in architecture, training data, and positional encoding schemes.

The layers that exhibit the lowest average curvature are also the ones where contextual curvature at individual tokens is the most predictive of next-token entropy, and these occur consistently in the middle of the network (Fig. 2). In GPT-2 XL, average curvature decreases steadily from early layers to a minimum around layer 23 (out of 48) (Fig. 2A). This reduction is accompanied by a monotonic increase in predictive correlation between curvature and next-token entropy, with maximal predictive power occurring near the layer of minimal curvature (Fig. 2C). The same pattern is observed in the Pythia-2.8B model (Fig. 2B, D), indicating that this relationship generalizes across models.

Although the resulting correlations are modest in absolute magnitude (peaking around $r \approx 0.15$), we suspected that this could be a result of how we defined our measure. Contextual curvature is measured at individual token positions and is a single scalar summary of a rich, high-dimensional representation. Predictive entropy depends on many additional factors encoded in the high-dimensional representation. Nevertheless, these correlations are highly consistent across layers, datasets, and model classes, and at the straightest layers exceed those obtained using alternative scalar features. By contrast, control measures based on activation magnitude and trajectory distance show weaker or later-emerging relationships with entropy (Fig. A1), indicating that curvature uniquely captures uncertainty-relevant geometry in the middle layers.

### 3.2. Training Links Curvature with Entropy

Previous work has established that curvature reduction in middle layers emerges gradually over the course of pre-training (Hosseini & Fedorenko, 2023; Skean et al., 2025). Here we asked whether the coupling between curvature and entropy emerges in tandem. Pythia-2.8B provides model checkpoints throughout training, allowing us to track this relationship over time. Specifically, we analyzed model checkpoints at logarithmically spaced intervals over a 300-billion-

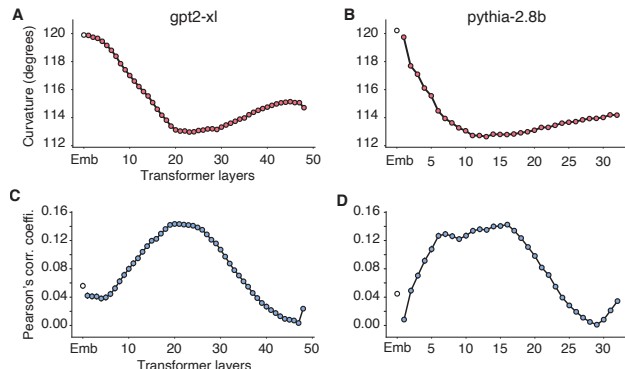

*Figure 2.* **Curvature and entropy relationship across layers.** (**A**) Average contextual curvature across transformer layers in GPT-2 XL on the long-context dataset. (**B**) Same as (A) for Pythia-2.8B. In both models, curvature decreases from early layers and reaches a minimum in the middle of the network. (**C**) Predictive performance (Pearson $r$) of contextual curvature for next-token entropy across layers in GPT-2 XL. (**D**) Same as (C) for Pythia-2.8B. In both cases, predictivity increases across layers and peaks around the layers of minimal curvature.

token training run $(0\%, 0.007\%, 0.07\%, 0.7\%, 7\%, 70\%)$, measuring internal curvature, its predictivity of next-token entropy, and average token-level entropy on the long-context dataset.

The representational curvature gradually decreases over all layers during the course of training (Fig. 3A). Initially, the layer-wise curvature profile is relatively flat, with all layers exhibiting high curvature comparable to the embedding layer (Fig. 3A; $0\%, 0.007\%, 0.07\%$). With additional training, curvature begins to decrease in early layers and reaches a minimum in the middle layers. By $70\%$ of training (Fig. 3A; $70\%$), the curvature profile closely resembles that of the fully trained model.

Critically, these changes in representational curvature coincide with the emergence of a link to output entropy. In the earliest stages of training, curvature is only weakly predictive of next-token entropy (Fig. 3B; $0\%, 0.007\%, 0.07\%$). However, around $0.7\%$ of training, a distinct transition occurs: curvature decreases substantially across the early-to-middle layers, and the predictive power of curvature rises sharply in tandem. This relationship continues to strengthen over training. By the final checkpoint, the middle layers are also the most predictive of entropy. The parallel training-dynamics analysis for control measures (activation magnitude and trajectory distance) shows weaker or later-emerging coupling, mirroring the static comparison (Fig. A3).

### 3.3. Trajectory-Aligned Perturbations Modulate Entropy

To investigate the extent to which curvature directly shapes uncertainty, we perturbed internal activations and measured

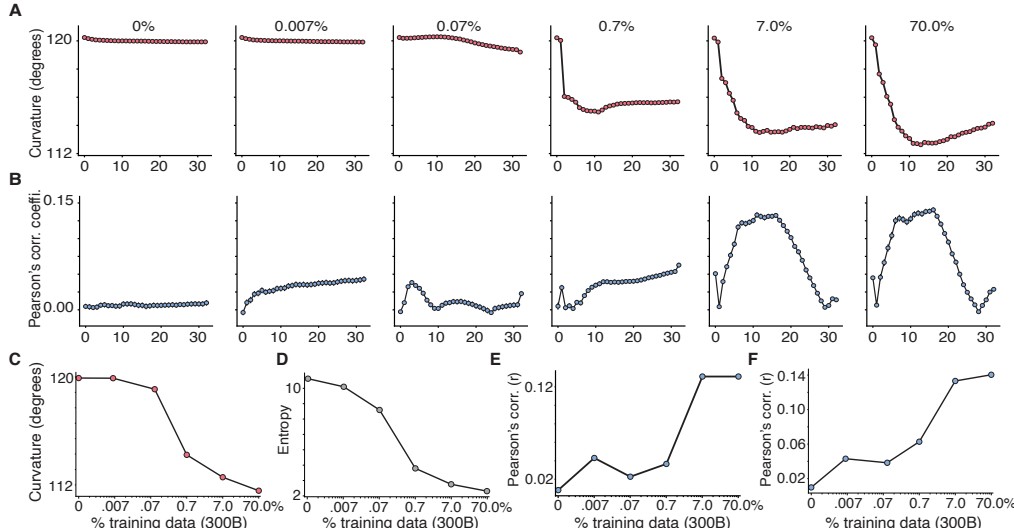

*Figure 3*. **Training dynamics of curvature and uncertainty.** (**A**) Layer-wise contextual curvature across training checkpoints for Pythia-2.8B on the long-context dataset. (**B**) Predictive performance (Pearson $r$) of contextual curvature for next-token entropy across layers at the same checkpoints. (**C**) Average curvature at the minimum-curvature layer over training. (**D**) Average token-level entropy over training. (**E**) Predictivity of contextual curvature measured at the minimum-curvature layer across checkpoints. (**F**) Predictivity of contextual curvature measured at the maximum-predictivity layer across checkpoints. Training progressively straightens internal trajectories and strengthens the coupling between curvature and predictive uncertainty.

the resulting changes in next-token entropy. Holding the input fixed lets us better isolate the effect of curvature on entropy, while varying the geometric subspace from which perturbations are drawn lets us test a sharper prediction of the temporal straightening hypothesis: entropy should respond to curvature changes along the model's representational trajectory—the direction along which extrapolation would occur—but not to curvature changes in arbitrary directions of activation space.

All interventions targeted middle layers, where straightening is most pronounced. For each token, we computed the induced change in curvature ($\Delta C$) and the corresponding change in next-token entropy ($\Delta H$). We used five perturbation types, progressively moving from unconstrained perturbations of activation space to perturbations constrained to subspaces aligned with the trajectory (Fig. 4A): (1) **full-space**: the entire activation space, with no geometric constraint; (2) **random-subspace**: a randomly oriented low-dimensional subspace, controlling for dimensionality alone; (3) **activation-subspace**: the top principal components of residual-stream activations, capturing data-aligned but trajectory-agnostic directions; (4) **trajectory-subspace**: the top principal components of each sample's token-to-token displacement vectors, aligning perturbations with the model's recent representational path; and (5) **planar-subspace**: the plane spanned by the two most recent displacements, the tightest possible alignment with the geometry in which curvature is defined (see Section 2.6 for a more detailed explanation).

We find that changes to curvature only produce a reliable effect on entropy when perturbations are restricted to the trajectory (Fig. 4). Specifically, perturbations drawn from the full activation space, randomly oriented low-dimensional subspaces, and activation-derived subspaces all failed to produce a reliable relationship between $\Delta C$ and $\Delta H$, suggesting that the model's predictive computations are robust to curvature changes outside of the trajectory. On the other hand, interventions restricted to the trajectory subspace yielded a robust curvature-entropy relationship, while the planar perturbations produced the largest effect size.

Thus, modifying curvature along directions in which the representation naturally evolves produces consistent directional changes in output entropy. These findings suggest that trajectory curvature is a behaviorally relevant geometric feature.

### 3.4. Curvature Regularization During Training Reduces Entropy

Finally, we explored whether insights about the link between representation and behavior can be leveraged during training to improve model reliability. We added an auxiliary loss that penalized mean curvature over each sequence (Fig. 5A) and trained a GPT-2 style model under three conditions, giving three models: baseline (next-token prediction only), *untangled* (curvature penalty encouraging straighter trajectories), and *tangled* (reversed penalty encouraging higher curvature).

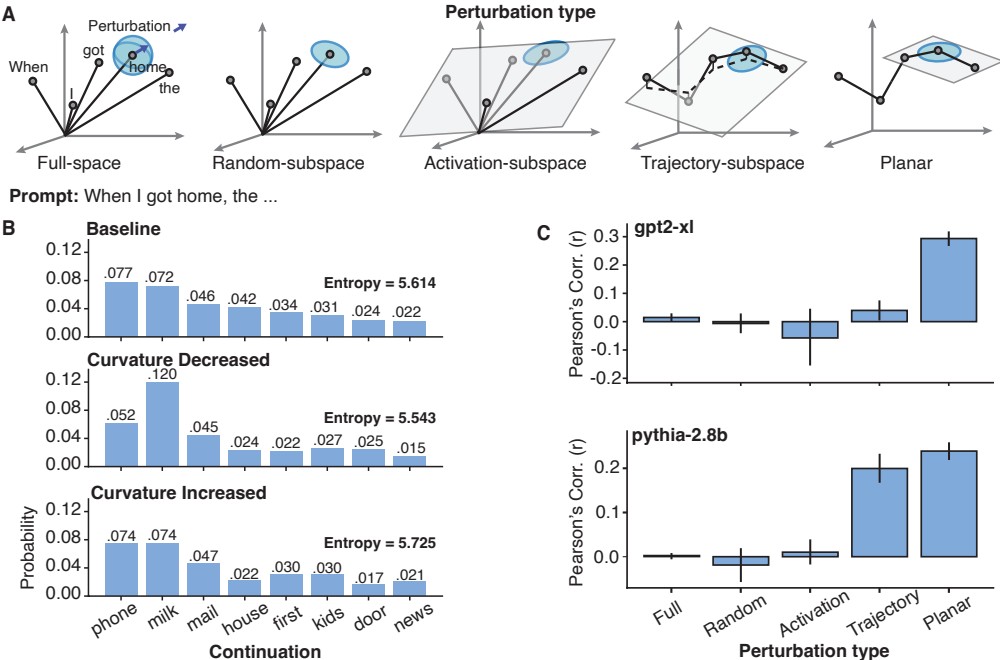

*Figure 4.* **Trajectory-aligned perturbations selectively couple curvature to uncertainty.** **(A)** Schematic of the five perturbation types, ranging from unconstrained to trajectory-aligned: *full-space* (entire activation space), *random-subspace* (random low-dimensional subspace), *activation-subspace* (top PCs of activations), *trajectory-subspace* (top PCs of token-to-token displacement vectors), and *planar-subspace* (plane of the two most recent displacements). **(B)** Example output distributions after perturbation at the token "the" in the prompt "When I got home, the ". One perturbation increases both curvature and entropy; the other decreases both. **(C)** Correlation between changes in curvature ($\Delta C$) and changes in entropy ($\Delta H$) induced by different perturbation families (bars: mean across samples; error bars: 95% CIs). Perturbation layer is 21/48 for GPT-2 XL and 11/32 for Pythia-2.8B.

We first verified that the auxiliary loss influenced representations as intended. Compared to baseline, untangled models showed lower curvature in middle layers, while tangled models showed higher curvature (Fig. 5C). Validation loss was similar across conditions (Fig. 5C, inset; Fig. A6), indicating that the regularization modulated geometry without degrading overall predictive performance. This could be due to the fact that cross-entropy loss primarily constrains the probability of the target token, allowing a degree of freedom in how the remaining probability mass is distributed across other tokens.

We found that models trained to untangle their neural trajectories showed lower token-level entropy across different validation sets compared to baseline models trained without the auxiliary loss, and the reverse was true for tangled models (Fig. 5D). While these effects were small, they align well with our predictions and generalize across multiple datasets. Several factors may contribute to the modest effect size, including the limited scale of both the model and the training dataset.

## 4. Discussion

We investigated whether contextual curvature—a geometric measure of how sharply representations bend over recent context—modulates next-token entropy in LLMs. Curvature predicts next-token entropy most strongly in middle layers, where trajectories are straightest. This relationship emerges over training, and perturbation experiments suggest entropy is selectively dependent on curvature: trajectory-aligned perturbations that increase curvature reliably increase entropy, while misaligned perturbations do not. Finally, we find evidence that curvature regularization can modestly influence entropy without degrading validation loss.

The temporal straightening hypothesis in LLMs posits that models transform their internal representations of input sequences into straighter trajectories to facilitate prediction via extrapolation (Hosseini & Fedorenko, 2023). We provide evidence for a natural implication of this hypothesis: trajectories with low curvature are easier to extrapolate from, yielding lower entropy output distributions. Combined with previous work, our observations suggest that in order to do next-token prediction, LLMs implicitly learn to straighten their internal trajectory over sequences of tokens. The localization of this effect to intermediate layers aligns with broader observations that middle layers are disproportionately informative for downstream behavior (Skean et al., 2025; Marshall & Kirchner, 2024; Bigelow et al., 2025; Lubana et al., 2025; Park et al., 2025).

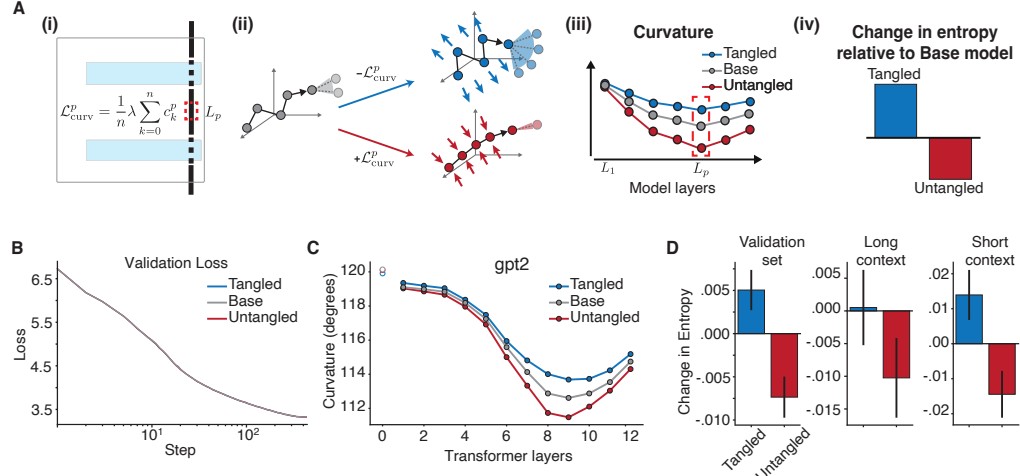

*Figure 5.* **(A)** Experimental design: (i) regularization is applied to the residual stream to modulate curvature, (ii) increasing curvature (Tangled) or decreasing it (Untangled), with (iii) hypothesized layer-wise curvature and (iv) hypothesized entropy effects. **(B)** Training loss across conditions, showing similar convergence. **(C)** Layer-wise curvature on the validation set. As hypothesized, the Untangled model shows lower curvature and the Tangled model shows higher curvature in middle layers. **(D)** Change in token-level entropy relative to baseline across three datasets. The Untangled model reduces entropy across all datasets; the Tangled model increases entropy in two of three.

Our findings additionally have implications for work in mechanistic interpretability. Recent activation-steering methods show that internal representations can be manipulated to control model behavior, including expressed uncertainty (Turner et al., 2024; Marks & Tegmark, 2024; Li et al., 2024; Rahn et al., 2024). Our perturbation analysis complements these methods by targeting the temporal geometry of representations: modifying how representations evolve across tokens, not just their state at a single position, also yields predictable behavioral changes.

Similarly, Lubana et al. (2025) argue that single-position methods such as sparse autoencoders miss the temporal structure of LM representations, and propose a decomposition that separates each token's representation into a predictable component and a novel component. They find that the predictable component traces smooth trajectories across tokens, which they identify as a form of temporal straightening. Counterfactual edits to this predictable component shift the model's next-token predictions in correspondingly different directions, paralleling our perturbation results. Huang et al. (2026) formalize a closely related principle, the Geodesic Hypothesis, positing that token trajectories trace locally linear geodesics on a semantic manifold, and show that an auxiliary training loss enforcing this structure improves data efficiency. Their loss is a manifold-aware analog of our curvature regularization; both auxiliary objectives constrain across-token trajectory geometry during training, with measurable functional benefits. Together, these lines of work point to across-token representational geometry as both functionally consequential and amenable

to targeted intervention, marking it as a promising target for mechanistic interpretation and behavioral control.

**Limitations and Future Directions** Our approach simplifies both the representations and the behavior of the models. Temporal straightening implies that the model is shaping the geometry of the representational manifold to make trajectories easier to extrapolate from. Our curvature measure captures one aspect of this, but predictability on a manifold is a richer problem: it also depends on whether nearby trajectories are converging toward similar predictions or diverging toward different ones, how many dimensions the trajectory occupies, and how quickly the representation moves through space. Contextual curvature reduces high-dimensional trajectories to a single scalar over a local window of points. Geometric descriptors that allow more flexibility might capture aspects of representation structure that curvature misses. Similarly, entropy quantifies uncertainty but not its exact structure. For example, output distributions could have comparable entropy, yet spread their probability mass across semantically similar tokens ('happy', 'glad') or dissimilar ones ('happy', 'expired'). Characterizing what geometric motifs correspond to different distributional structures—not just overall uncertainty—would further clarify the link between geometry and behavior.

We used small-scale LLMs for the training experiments. Whether the curvature-entropy relationship holds in larger foundation models, different architectures, or multi-modal systems remains untested, as does generalization to more diverse datasets. Also, while we show that curvature regular-

ization reduces entropy, whether this translates to improved calibration, robustness, or downstream task performance is an open question.

Our experiments indicate that geometric regularization can modestly influence model confidence: nudging the representations towards straighter trajectories reduces token entropy without degrading validation loss, while penalizing straightness increases entropy. These effects were small. We believe this could be due to the limited scale of our models and training data, a hypothesis that remains to be tested. This perspective is supported by recent work with larger models, showing that uncertainty and calibration can be shaped through auxiliary training objectives (Krishnan et al., 2024), as well as with evidence that regularizing representation structure can improve learning (Bardes et al., 2022; Li et al., 2025; Huang et al., 2026).

## Impact Statement

This paper presents work whose goal is to advance the understanding of how large language models represent and process sequential information. We identify geometric properties of internal representations that relate to model uncertainty, which may inform future efforts to improve model calibration and interpretability. We do not foresee specific negative societal consequences arising from this work.

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

## A. Appendix

### A.1. Additional Representation Measures

**Contextual Magnitude**: to estimate representational magnitude, we computed the L2 norm for each token embedding at all layers. For a given token $x_k^p$, its magnitude is:

$$\mu_k^p = \|x_k^p\|_2 = \sqrt{\sum_{j=1}^{d} (x_{k,j}^p)^2}$$

where $d$ is the dimensionality of the hidden state. We then averaged this value over a backward-looking window to obtain the contextual magnitude:

$$M_k^p = \frac{1}{3} \sum_{i=k-3}^{k-1} \mu_i^p$$

This feature captures the overall size of the representation vectors leading up to each token.

**Contextual Distance**: we defined token distance as the Euclidean norm of the trajectory segment immediately preceding each token. That is, for token $x_k^p$, the contextual distance is:

$$d_k^p = \|x_{k+1}^p - x_k^p\|$$

Analogous to the previous measures, we smoothed this distance over a local window to define contextual distance as:

$$D_k^p = \frac{1}{3} \sum_{i=k-3}^{k-1} d_i^p$$

This measure captures how quickly the model's representation is moving through space, without reference to its direction or curvature.

### A.2. Additional Regression Analysis

**Regression Analysis with Unigram Probability Control**: Because unigram probability is inherently predictive of next-token entropy, its influence may be partially reflected in the contextual representation itself. That is, if some of the representation's predictive power stems from encoding token frequency, then any observed relationship between representational geometry and entropy could be confounded by this prior information. To isolate the genuinely contextual contribution of the representation, we therefore control for unigram probability—allowing us to assess whether geometric features like curvature explain additional variance in entropy beyond what is attributable to token frequency alone.

To determine whether contextual curvature explains variance in entropy beyond that captured by unigram probability, we performed a model comparison analysis. Specifically, we fit two linear regression models for each fold of 10-fold cross-validation: a baseline model using unigram probability alone

as a predictor, and a full model including both unigram probability and contextual curvature.

For each fold, we computed the Pearson correlation ($r$) between predicted and observed entropy for both models, and recorded the difference in $r$ (full model minus baseline). This yielded a distribution of 10 correlation differences across folds.

We report the mean increase in $r$ attributable to adding curvature as the main result. As with previous analyses, we computed the 95% confidence interval of the correlation differences using the $t$-distribution with 9 degrees of freedom. This procedure quantifies whether contextual curvature provides a consistent and statistically reliable improvement in predicting model uncertainty beyond unigram probability.

### A.3. Calculating Unigram Probabilities

To estimate each model's unigram token probabilities, we approximated the marginal distribution over tokens by sampling from the model's own generative distribution. Specifically, we generated 1,000 short sequences (30 tokens each) using nucleus sampling (top $k = 1000$), starting from the beginning-of-sequence token. For each generated sequence, we recorded the predicted token probability distribution at every time step (i.e., the model's softmax over the vocabulary at each position).

Let $p_t(v)$ denote the predicted probability of vocabulary item $v$ at position $t$ in a sampled sequence. We accumulated these across all positions and samples to estimate the marginal probability of each token:

$$\hat{P}(v) = \frac{1}{T} \sum_{i=1}^{N} \sum_{t=1}^{L_i} p_t^{(i)}(v)$$

where $N$ is the number of sampled sequences, $L_i$ is the length of the $i$-th sequence, and $T = \sum_{i=1}^{N} L_i$ is the total number of tokens across all sequences.

This yielded an empirical approximation of the model's unigram distribution, which was normalized to ensure it summed to one. These estimated unigram probabilities were then used as covariates in subsequent regression analyses to control for frequency-based effects on next-token entropy.

### A.4. Importance Reweighting for Perturbation Correlations

Different perturbation families can produce systematically different distributions of curvature change $|\Delta C|$, which could confound comparisons of the $\Delta C$–$\Delta H$ correlation across families. To ensure that observed differences in correlation reflect geometric alignment rather than distributional mismatch, we apply importance reweighting to equalize

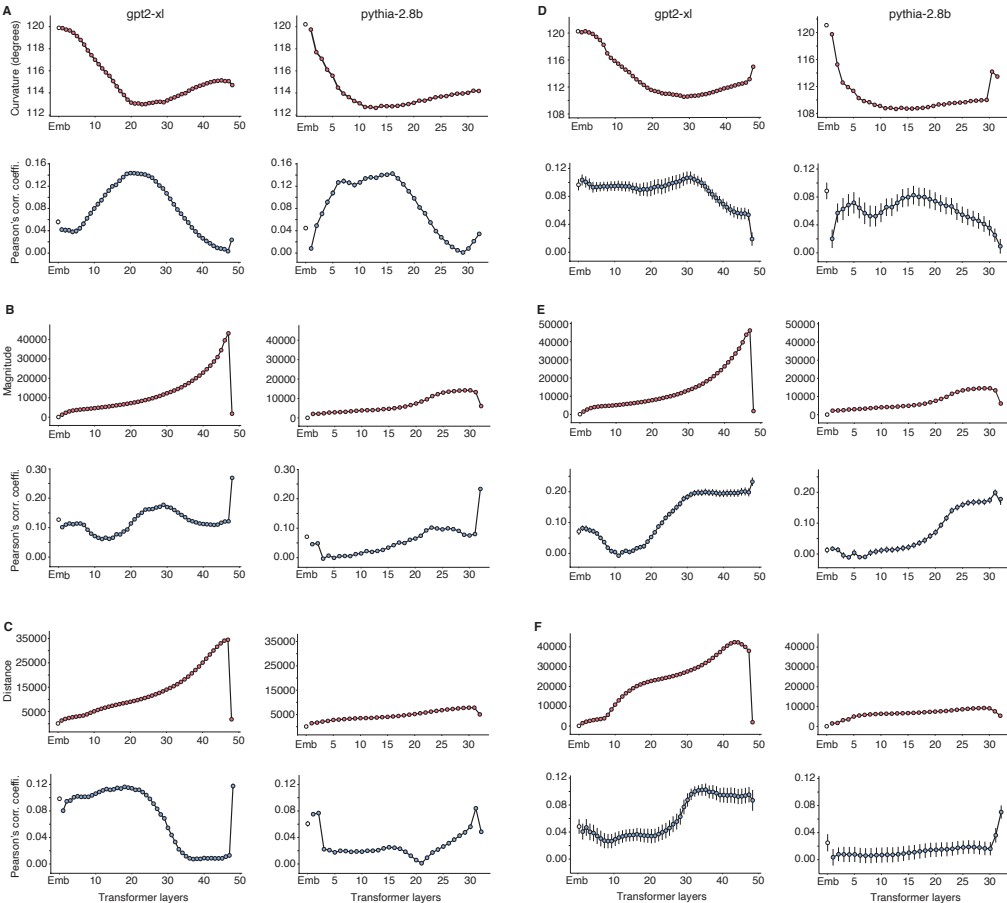

*Figure A1.* **Predicting next-token entropy with three geometric measures.** For curvature (A,D), magnitude (B,E), and distance (C,F), we plot the measure values across layers (red) and the Pearson correlation between predicted and true next-token entropy (blue). Left column: long-context dataset; right column: short-context dataset.

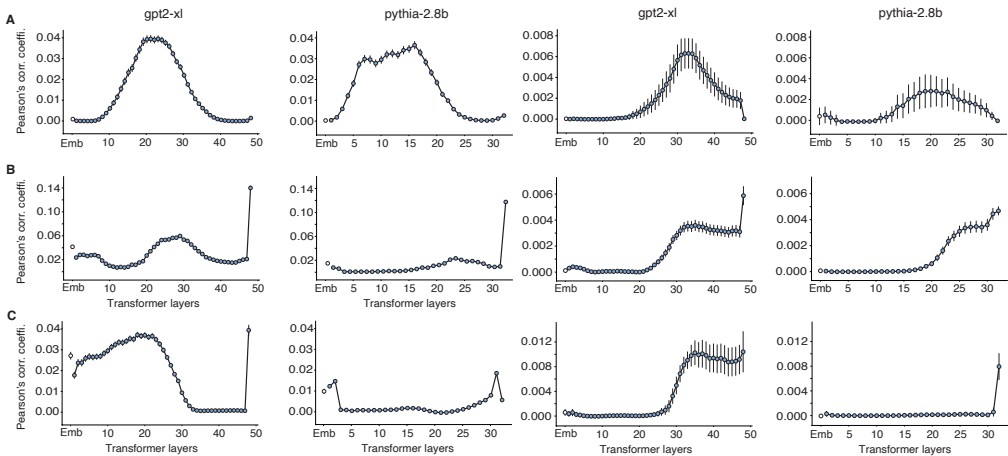

*Figure A2.* **Controlling for unigram probability in predicting next-token entropy.** Plots show the difference in Pearson correlation between models that include curvature and those that use unigram probability alone. Error bars indicate 95% confidence intervals. (A) Curvature, (B) Magnitude, (C) Distance. Left two columns: long-context dataset; right two columns: short-context dataset.

the marginal distribution of $|\Delta C|$ across perturbation types before computing correlations.

We first construct a reference distribution by pooling $|\Delta C|$ values from the full-space perturbation across all layers and binning them into a histogram with $B = 100$ equal-width

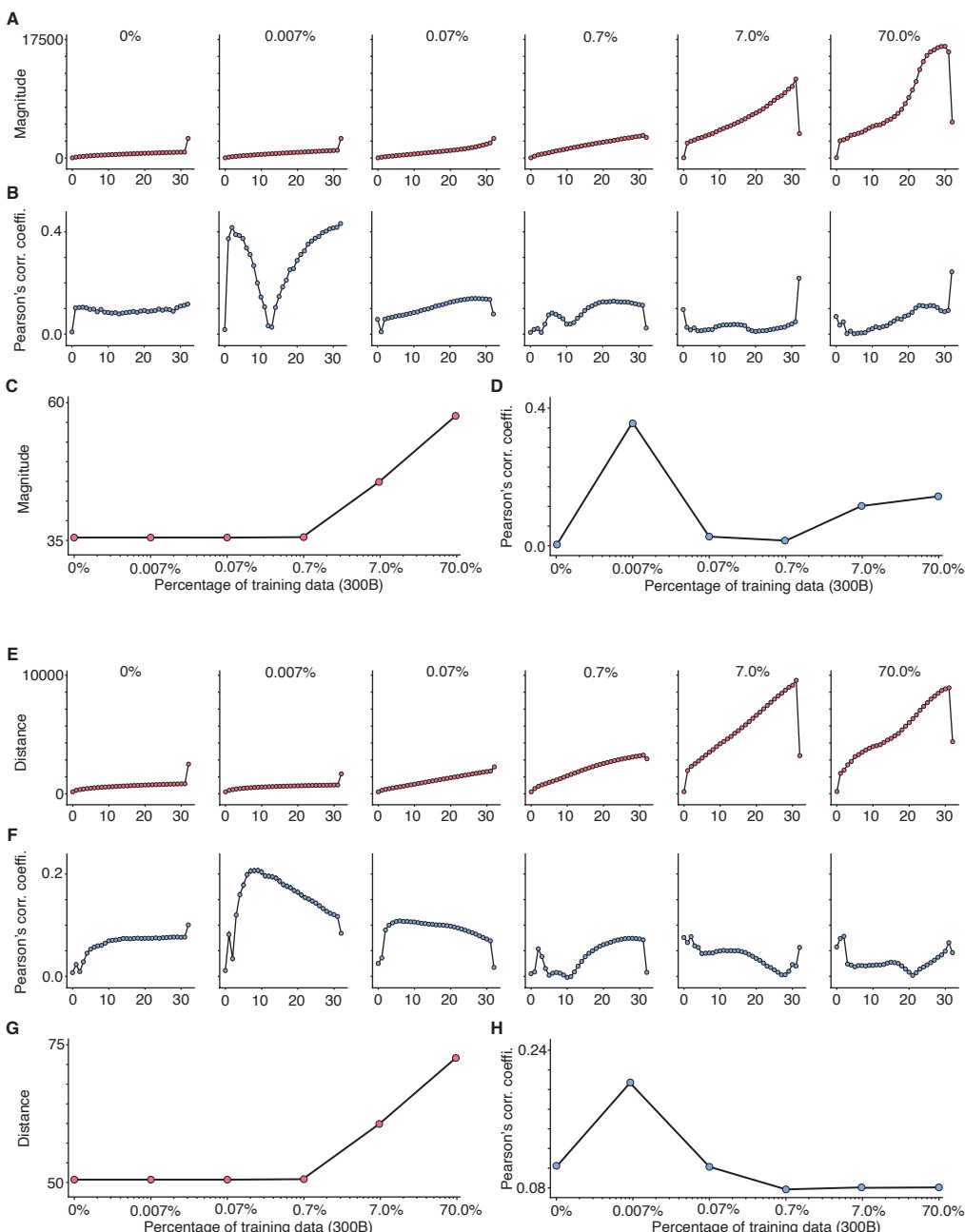

*Figure A3.* **Training dynamics for magnitude and distance in the long-context dataset.** (A) Magnitude across layers over training. (B) Correlation between magnitude and entropy over training. (C) Minimum magnitude at each checkpoint. (D) Maximum correlation at each checkpoint. (E–H) Parallel analyses for distance.

bins. Let $q_b$ denote the proportion of reference samples falling in bin $b$. For each perturbation family, we compute the analogous histogram $p_b$ from that family's $|\Delta C|$ values and assign each perturbation $i$ falling in bin $b$ the importance weight

$$w_i = \frac{q_{b(i)}}{p_{b(i)} + \epsilon},$$

where $\epsilon = 10^{-12}$ prevents division by zero. Weights are

clamped to $[\frac{1}{\tau}, \tau]$ with $\tau = 10$ to limit the influence of any single sample.

The per-token correlation between $\Delta C$ and $\Delta H$ is then computed as a weighted Pearson correlation using the importance weights $w_i$. These per-token correlations are averaged across tokens, and 95% confidence intervals are obtained by bootstrap resampling over tokens with $B_{\text{boot}} = 2{,}000$ replicates and Fisher $z$-transformation.

## A.5. Perturbation Effects Across Layers

The perturbation experiments reported in the main text were conducted in intermediate layers where we expect the trajectory geometry to be implicated in model behavior. Here we examine how the $\Delta C$–$\Delta H$ relationship changes when perturbations are applied at early and late layers.

At early layers, the relationship between $\Delta C$ and $\Delta H$ is generally weak across all perturbation types (Fig. A4A). This is consistent with the observation that curvature-entropy coupling is weak at these layers (Fig. 2).

At late layers in Pythia-2.8B, trajectory-aligned perturbations–particularly planar perturbations–produce significant *negative* $\Delta C$–$\Delta H$ correlations, reversing the pattern observed at middle layers (Fig. A4A). To understand this reversal, we examined the Spearman rank correlation between the perturbed and unperturbed output distributions for each perturbation type across layers (Fig. A4B). At middle layers, all perturbation types largely preserve the output ranking, indicating that the interventions modulate uncertainty without fundamentally altering the model's predictions. At late layers, however, trajectory-aligned perturbations substantially degrade the output ranking, suggesting that these interventions are no longer selectively modulating uncertainty but are instead disrupting the model's output distribution. The resulting negative correlations are therefore difficult to interpret as evidence of a meaningful geometric relationship: a perturbation that scrambles the token ranking and happens to compress the output distribution does not reflect the same mechanism as one that shifts entropy while preserving the structure of the prediction. This motivates our focus on the middle layers, where perturbations modulate curvature and entropy while leaving the output distribution largely intact.

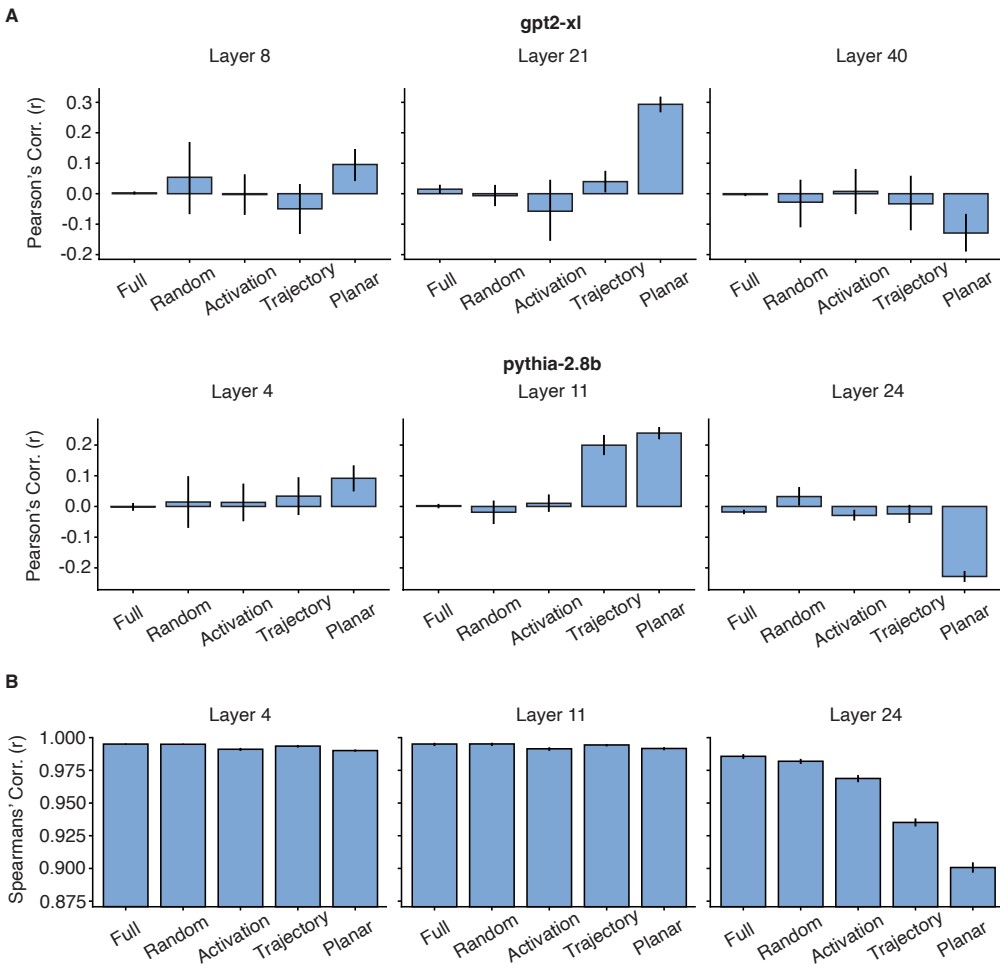

*Figure A4.* $\Delta C$–$\Delta H$ **correlations across layers.** **(A)** Correlation between induced curvature change and entropy change for each perturbation type at early, middle, and late layers. Bars show mean per-token correlation; error bars show 95% bootstrap CIs. **(B)** Spearman rank correlation between perturbed and unperturbed output logit distributions for each perturbation type at early, middle, and late layers for Pythia-2.8B. At early and middle layers, all perturbation types largely preserve the output ranking. At late layers, trajectory-aligned perturbations substantially degrade it, indicating nonspecific disruption rather than selective modulation of uncertainty.

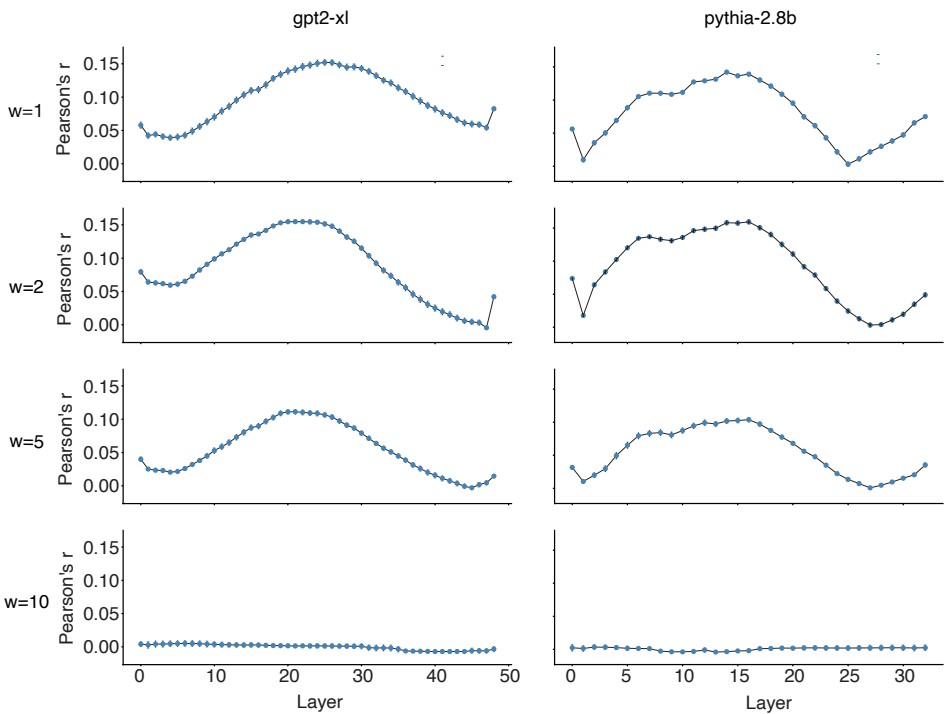

*Figure A5.* **Window size sweep.** Predictive performance (Pearson r) of contextual curvature for next-token entropy across layers for window sizes w = 1, 2, 5, 10 in GPT-2 XL (left) and Pythia-2.8B (right). The middle-layer predictivity peak is stable across window sizes, with performance decreasing with larger window sizes at 5 and beyond.

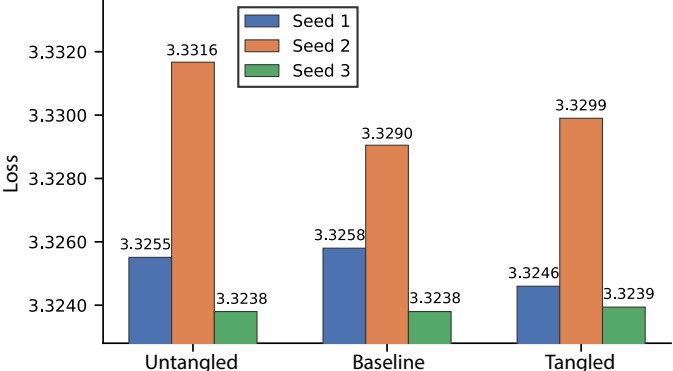

*Figure A6.* **Final validation loss across regularization types.** Final validation losses across the Baseline, Untangled, and Tangled models, demonstrating that losses are comparable.

