# OpenReview forum: "Representational Curvature Modulates Behavioral Uncertainty in Large Language Models"
_ICML.cc/2026/Conference — ICML 2026 regular_

### Official Review · Reviewer_WjcK · 2026-03-07

**Soundness:** 2
**Presentation:** 3
**Significance:** 2
**Originality:** 3
**Overall Recommendation:** 4
**Confidence:** 3

**Summary:**

The paper argues that straighter hidden-state trajectories in LLMs are associated with lower next-token uncertainty. It shows that this curvature–entropy link is strongest in middle layers, emerges during training, and can be partly controlled with perturbations or regularization, suggesting that trajectory straightening may play a functional role in prediction rather than being just a descriptive pattern.

**Compliance With Llm Reviewing Policy:**

Affirmed.

**Final Justification:**

The rebuttal addressed my concerns. The paper studied an interesting relation between curvature and next-token prediction entropy.

**Key Questions For Authors:**

* The regularization result makes curvature look less like a correlate of entropy and more like a direct control variable. If entropy changes while validation loss stays the same, why shouldn’t we read this as evidence that curvature is effectively determining uncertainty?


* The observation that middle layers are both straightest and most predictive of entropy sounds reminiscent of an information-bottleneck story, where intermediate representations compress context into a task-relevant predictive code. Do you see your result that way, or do you think “trajectory straightening” is a different phenomenon from compression in the information-bottleneck sense?

**Limitations:**

Yes, the authors have discussed the limitations.

**Strengths And Weaknesses:**

**Strengths:  **

1. This paper checks where the studied effect is strongest, how it emerges over training, and whether trajectory-aligned perturbations selectively change entropy while misaligned perturbations do not. That makes the mechanistic claim meaningfully stronger than a simple observational result.

**Weaknesses:**

1. The main variables in the study are scalars, ie, contextual curvature, computed over a local trajectory window, and next-token entropy. That makes the story elegant, but also lossy: very different geometric structures can share similar curvature, and very different output distributions can share similar entropy. So the paper may be capturing a real effect while still missing the richer mechanism underneath it.

2. The regularization experiments are done on small-scale LLMs, and the paper says outright that it remains untested whether the same relationship holds for larger foundation models, different architectures, multimodal systems, or more heterogeneous datasets. That matters a lot here, because the paper is close to proposing a general principle of autoregressive prediction. Right now, I think the evidence is good for “this happens in the studied setting,” but weaker for “this is a broad law of LLM computation.”

3. Even if curvature regularization reliably lowers entropy, the paper does not show that this improves calibration, robustness, task accuracy, or downstream usefulness. And because validation loss stays similar, one interpretation is that it mostly reshapes probability mass among wrong answers rather than improving decision quality.

---

> ### Author Rebuttal · Authors · 2026-03-31
>
> We want to thank the reviewer for their insightful comments and have included additional analysis based on their suggestions
>
> *Refer to Fbv8 for a comment on an error in the paper.*
>
> ### Curvature is a scalar:
> We agree. See “Effect sizes” in our Fbv8 reply for our response.
>
> ### Applicability of regularization results to all LLMs:
> We distinguish between the core findings and regularization experiments. Curvature-entropy correlation and causal specificity of trajectory-aligned perturbations are demonstrated across two architecturally distinct model families and both datasets, so the phenomenon itself generalizes beyond a single setting. The regularization experiments are more limited in scale, which we agree is a constraint. The reported regularization strength was the maximum possible without degrading validation loss, suggesting a tradeoff between geometric regularization and the primary objective. Larger models may be easier to regularize given greater capacity, supported by concurrent work (Huang et al., 2026) described below.
>
> ### Utility of regularization:
> We agree that we have not demonstrated downstream utility at our scale, and we are not claiming that lowering entropy is inherently useful. The point of our regularization experiment is to show that modifying representational geometry has a predictable, directionally consistent effect on model output, as predicted by the temporal straightening hypothesis, which provides further evidence that curvature is linked to behavior. For stronger evidence of whether this principle translates to practical gains at scale, we can turn to current work from Huang et al. 2026, which we think addresses the reviewer’s point. Their Semantic Tube regularizer, which similarly enforces trajectory straightness, yields substantial improvements in both post training task accuracy and data efficiency across six model families and up to 8B parameters. This suggests that the geometric relationship we identify can have practical implications at scale.
>
> As the reviewer suggested, we performed a preliminary analysis on the relationship between curvature and calibration, and found a degradation in calibration (how well accuracy tracks confidence) for sequences with high curvature. We specifically focused on Gemma-2-27B because the models we used in the paper do not perform well on MMLU, prohibiting us from computing calibration properly.
>
> *[Figure: curvature vs calibration](https://drive.google.com/file/d/1Q1w6B35wh1w0Tel2mHlCAA1Uw1RS0dBR/view?usp=sharing)*
>
> ### Does curvature control entropy?
> We appreciate this framing and agree the regularization result goes beyond correlation. However, we stop short of calling curvature a direct control variable for two reasons. First, the effect sizes are modest, suggesting one contributing factor among others. Second, the fact that entropy changes while validation loss doesn't is partly a consequence of the training objective itself: during pre-training cross-entropy loss is computed against one-hot targets, so it only reflects the probability assigned to the correct token, ignoring the distribution over the rest of the vocabulary. Thus, an entire dimension of output is left unconstrained. Curvature could serve as one geometric feature shaping the output.
>
> ### Information-bottleneck perspective
> We see these as closely related. The information bottleneck predicts that intermediate layers compress context into a task-relevant predictive code. To perform temporal prediction, LLMs similarly need to compress past information so that representations of preceding states predict future ones. Under the hypothesis that past states can linearly extrapolate to future ones, reducing curvature corresponds to discarding trajectory structure irrelevant to extrapolation, preserving only what is needed. Our perturbation results add specificity to this picture: they suggest what matters is not just that information is compressed, but that it is compressed specifically along the trajectory subspace, since perturbations in that subspace maximally affect behavior..
>
> The deeper question is why compression would take this particular geometric form. While information theory provides tools for measuring the relationship between past and future states via mutual information, it does not provide a mechanism for transforming past states to predict the future. The temporal straightening framework offers such an account: successive nonlinear transformations across layers yield latent states that are progressively more predictable via linear extrapolation. Under this view, middle-layer compression is specifically organized to make the trajectory easier to extrapolate from, which is why curvature, rather than other compression measures, is the geometric feature most tightly coupled to predictive uncertainty.
>
> ### References
> * Huang, H., LeCun, Y., & Balestriero, R. (2026). Semantic Tube Prediction: Beating LLM Data Efficiency with JEPA. arXiv preprint arXiv:2602.22617.

---

> > ### Author Rebuttal · Reviewer_WjcK · 2026-04-02
> >
> > I thank the authors for their rebuttal. My concerns are resolved.

---

> > > ### Author Response · Authors · 2026-04-08
> > >
> > > We’d like to thank the reviewer for their time, their constructive feedback throughout this process, and for confirming that their concerns are fully resolved. We appreciate their engagement with our work.

---

### Official Review · Reviewer_2idr · 2026-03-08

**Soundness:** 3
**Presentation:** 3
**Significance:** 2
**Originality:** 3
**Overall Recommendation:** 4
**Confidence:** 4

**Summary:**

The paper establishes a connection between contextual curvature and contextual entropy. Contextual curvature treats the sequence of latent representations at a given layer as a discretized curve and measures the degree to which it deviates from linearity. Contextual entropy, defined as the sum of next-token prediction entropies across all positions, serves as a measure of the model's overall predictive uncertainty. Through a series of experiments, the authors demonstrate that contextual entropy tends to increase with contextual curvature, and that this correlation holds consistently across training stages, model layers, and different model architectures. The experiments further reveal that contextual curvature decreases progressively through the early-to-middle layers before exhibiting a modest increase toward the final layers, suggesting that the middle layers are most predictive of contextual entropy.

**Compliance With Llm Reviewing Policy:**

Affirmed.

**Final Justification:**

I appreciate the effort of the authors to clarify ambiguous content in the rebuttal.

In the last reply, the authors have fully addressed one out of three concerns:
1. Contextual entropy is different from sentence surprisal and predictability is not transferable. Therefore, the finding here is indeed original.

Two concerns are partially addressed:
1. Regarding the extrapolation hypothesis, "the modest regression value reflects that curvature is a scalar compression of a high-dimensional process, not a failure of the extrapolation hypothesis." There is no justification for this claim; however, it makes sense to me that this has a high chance of occurring given the current results. Also, the intervention experiment shows that increased curvature leads to increased entropy, but is not particularly helpful as the effect is small.
2. "Further evidence that LLMs learn temporal straightening points to a convergence between biological and artificial systems." This shows the broader impact of this work on neuroscience; however, it lacks discussion of how this understanding could benefit LLM research, which is my main concern.

Given the above justifications, I have increased my originality, significance, and overall scores. I suggest the authors include the discussed content, specifically the explanation of contextual entropy and the perturbation experiment, in the final version.

**Key Questions For Authors:**

1. In [1], matrix-based entropy is presented as a more general metric that subsumes contextual curvature. If this is the case, could matrix-based entropy serve as a stronger predictor of contextual entropy? The authors are encouraged to include a direct comparison of the predictive effectiveness of contextual curvature versus matrix-based entropy from [1].
2. What theoretical account do the authors offer for the observed correlation between contextual curvature and contextual entropy? A more principled explanation would strengthen the paper's contribution.
3. What insights or potential applications from computational neuroscience might arise from interpreting the sequence of latent representations as undergoing curvature straightening over "time"  (which is model's layers in the paper context)?
4. The "long-context" datasets used in this work contain only a few hundred tokens, which falls within the small-to-medium range by current standards and is well within the comfortable operating range of most modern language models. It remains unclear how the reported findings generalize to substantially longer contexts , e.g. sequences exceeding 2,000 tokens. The authors should discuss or empirically investigate this regime.

**References**

[1] Skean, Oscar, et al. "Layer by Layer: Uncovering Hidden Representations in Language Models." *International Conference on Machine Learning*. PMLR, 2025.

**Limitations:**

Yes.

**Strengths And Weaknesses:**

### Strengths

1. The paper provides an explicit correlation between contextual curvature and contextual entropy, which offers additional empirical support for the linear representation hypothesis.
2. The experiments are thorough and substantiate the central claims. The proposed correlation is showed to exhibit across training stages, across model layers, and through controlled experiments that probe the causal relationship.

### Limitations

1. The contribution is somewhat limited in scope, as it primarily extends the surprisal analysis in [1] to a more general version - contextual entropy - with additional experiments, without introducing fundamentally novel theoretical or application insights.
2. The content is relatively sparse in analytical depth; the majority of the text is devoted to describing experimental procedures rather than offering meaningful interpretation or broader implications of the findings.
3. Minor errors:
   - The description of the LAMBADA dataset appears to be copied verbatim rather than paraphrased in the authors' own words.
   - Figure 1 is not referenced anywhere in the main text.
   - Wrong descriptions for Figure 4B and Figure 4C.

**References**

[1] Hosseini, Eghbal, and Evelina Fedorenko. "Large language models implicitly learn to straighten neural sentence trajectories to construct a predictive representation of natural language." *Advances in Neural Information Processing Systems* 36 (2023).

---

> ### Author Rebuttal · Authors · 2026-03-31
>
> We would like to thank the reviewer for the insightful comments and included additional analysis based on their suggestions.
>
> *Note to the reviewer: Refer to Fbv8 for comment on an error in the paper*
>
> ### Scope, Analysis, and Theory:
> To address concerns about scope, we will reframe the text to emphasize two key departures from prior work. First, we link internal representations directly to model behavior rather than input encoding. Second, instead of averaging surprisal over entire sequences (Hosseini, 2023), we link curvature to next-token entropy at individual positions, enabling targeted perturbations (Figure 4) beyond correlation (Figures 2, 3).
>
> We agree that our current setup prioritizes empirical validation over analytical depth. Our approach is to first empirically establish a theoretically motivated hypothesis., which we expand on here. By design, LLMs must represent past information in order to predict the future. While information theory provides a toolkit for measuring this relationship via mutual information, it does not provide a mechanistic account of how past states are used to predict the future. The Kalman filter provides one such account, but it is defined over linear dynamics, while video and language are characterized by non-linear relationships. To provide a functional account of prediction in these domains, the work of Hénaff (2018 NYU thesis, 2019 Nature Neuroscience) and Hosseini (2023, NeurIPS) assume that the successive non-linear transformations of the input—such as across Transformer blocks—yield latent states that are predictive of future states via simple linear extrapolation. Thus, the degree to which extrapolation captures true future states should be directly reflected in the behavioral entropy of the model. This theoretical link is the primary reason we chose to compare curvature and entropy.
>
> This perspective is underscored by Huang et al. (2026), who propose optimal hidden-state trajectories should be locally linear. Citing curvature straightening (Hosseini & Fedorenko, 2023), they introduce a regularizer that yields substantial efficiency and accuracy gains across various models (Llama3, Gemma2, etc.) and environments. Our findings are complementary: they show enforcing straightness improves post-training, while we provide a mechanistic account for LLMs trained purely on next-token prediction.
>
> ### Matrix-based Entropy:
> *[Figure: ME and entropy](https://drive.google.com/file/d/1QRg1BcBm3cILTJjjHmtJnGlU2vss3nby/view?usp=sharing)*
>
> Skean et al. (2025) qualitatively link trajectory curvature to skewed eigenvalue spectra, but their theorems lack a formal link to curvature. Matrix-based entropy measures the spread of representations independent of token ordering, whereas contextual curvature captures sequential directional change. Our new results show matrix-based entropy underperforms contextual curvature in predicting next-token entropy and lacks the middle-layer predictivity peak. We plan to formalize this relationship in future work.
>
> ### Computational Neuroscience:
> Our work is directly motivated by computational neuroscience. Hénaff et al. (2021) showed temporal straightening in the primate visual cortex, and recent fMRI work demonstrates similar straightening of human language representations (Xu et al., 2025).
>
> ### Extra Long Context:
> *[Figure: curvature entropy in long context](https://drive.google.com/file/d/1VWu3kG6ZovgxkFoZ7TBYwJwfApC92ykP/view?usp=sharing)*
>
> We used Claude Opus 4.6 to generate 10 narrative passages averaging 1800 tokens (Pythia context-window is 2048 tokens), each constructed to maintain a sustained contextual thread requiring integration of information across the full passage for accurate next-token prediction. The curvature-entropy relationship in this dataset follows the same pattern observed in the paper: curvature decreases to a minimum in middle layers, and predictivity of next-token entropy peaks at these same layers.
>
> ### References:
> *Huang, H., LeCun, Y., & Balestriero, R. (2026). Semantic Tube Prediction: Beating LLM Data Efficiency with JEPA. arXiv preprint arXiv:2602.22617.
>
> *Jiaming Xu, Jerry Tang, Alexander Huth, Robbie Goris, 2025, “Temporal Straightening as a Predictive Mechanism in Human Language Processing”, Computational Cognitive Neuroscience (CCN)
>
> *Hénaff, Olivier J. 2018. “Testing a Mechanism for Temporal Prediction in Perceptual, Neural, and Machine Representations.”, Doctoral Thesis, New York University
>
> *Hénaff, Olivier J., Yoon Bai, Julie A. Charlton, Ian Nauhaus, Eero P. Simoncelli, and Robbe L. T. Goris. 2021. “Primary Visual Cortex Straightens Natural Video Trajectories.” Nature Communications 12 (1): 5982.

---

> > ### Author Rebuttal · Reviewer_2idr · 2026-04-02
> >
> > Thank you for your reply.
> >
> > Some of my concerns have been addressed:
> > 1. Matrix-based entropy does not show a correlation with contextual entropy, which suggests the richer expressiveness of contextual curvature.
> > 2. The link between contextual curvature and contextual entropy generalizes to longer context lengths.
> >
> > However, most of my main concerns remain partially resolved, or not at all. I list them below:
> > 1. The contribution is limited to generalizing the surprisal into contextual entropy. The authors respond that the paper (1) measures next-token entropy at individual positions, instead of averaging surprisal over entire sequences as in Hosseini and Fedorenko (2023), and (2) enables targeted perturbations beyond correlation. I agree with (1) that individual positions are indeed a new contribution. However, this contribution is not significant and is predictable from Hosseini and Fedorenko (2023), as contextual entropy at position $i$ could be considered as an average entropy (generalized surprisal) of the sequence of length $i$. Regarding (2), the targeted perturbations indeed verify the correlation again, but do not clearly show what exactly the correlation is (does increased curvature increase entropy, or the opposite?) or how applications could take advantage of that correlation (e.g., more stable generation).
> > 2. The paper lacks theoretical motivation. The authors respond by stating the theoretically motivated hypothesis that latent states are predictive of future output states via simple linear extrapolation. However, the experiment using linear regression, despite showing a consistent correlation between contextual curvature and contextual entropy, does not achieve high absolute correlation values (0.04–0.16 Pearson correlation score in Figures 2 and 3). This reflects the fact that latent states are indeed not predictive via simple linear extrapolation, but must be via a more complex relation.
> > 3. The paper lacks insights or potential applications from computational neuroscience. The authors respond with a link to two works in neuroscience that observe similar behaviour in the human brain. However, the insights or potential applications from these observations are not explicitly stated, which does not resolve my concern.
> >
> > Given the above comments, I decide to keep my original score.
> >
> > **References**
> >
> > Hosseini, E., & Fedorenko, E. (2023). Large language models implicitly learn to straighten neural sentence trajectories to construct a predictive representation of natural language. Advances in Neural Information Processing Systems 36.

---

> > > ### Author Response · Authors · 2026-04-08
> > >
> > > We thank the reviewer for their continued engagement, and we appreciate the opportunity to clarify the theoretical distinctions and highlight how our findings build upon prior work.
> > >
> > > ### Regarding the connection to Hosseini & Fedorenko (2023):
> > > We understand why the conceptual link between surprisal and entropy might create the appearance that our findings are a direct corollary. However, there are fundamental mathematical and mechanistic distinctions that separate our token-level behavioral link from their sequence-level input analysis.
> > >
> > > **Entropy is an Output and Token-Level Metric:**
> > >
> > > The prior work relates internal geometry to properties of the input — surprisal is defined by the identity of the token that was actually inputted. Moreover, their analysis operates at the sequence level, correlating average curvature with average surprisal. Our work establishes a token-level link between internal geometry and the model's output distribution.
> > >
> > > **Entropy is Not Generalized Surprisal:**
> > >
> > > To clarify our metric: contextual entropy at position $i$ is the entropy of the single next-token distribution at that position, rather than an average over prior positions.
> > >
> > > We believe the reviewer is suggesting that because entropy is expected surprisal, a correlation between average curvature and average surprisal should imply a correlation between average curvature and entropy. However, this implication does not necessarily hold. Consider a model where the entropy of the output distribution is the same at every position in every context, but the probability mass is distributed over different tokens depending on context. Entropy is constant, so Corr(C, H) = 0. Yet surprisal still varies — sometimes the drawn token is a likely one, sometimes not — and unlikely draws plausibly increase downstream curvature. So Corr(C, S) can be positive while Corr(C, H) = 0.
> > >
> > > **Perturbation Experiments Are Directional and Causal:**
> > >
> > > Our perturbation experiments directly measure the correlation between $\Delta C$ and $\Delta H$ following an intervention. Because the $r$ value for trajectory-subspace perturbations is positive, the data explicitly demonstrates the directionality: increased curvature leads to increased entropy.
> > >
> > > We emphasize that these experiments are causal interventions designed to move beyond observational correlations. While observational curvature and entropy are both influenced by the input (introducing potential confounds), our perturbations hold the input fixed while directly manipulating the internal representation. The subspace design further isolates the representational trajectory specifically: even when perturbation types are matched for the magnitude of curvature change (Figure A4), only trajectory-aligned perturbations produce systematic entropy changes.
> > >
> > > ### Objective
> > > Ultimately, the primary objective of our perturbation experiments is not to immediately optimize downstream performance, but rather to rigorously test the temporal straightening hypothesis. We believe that advancing our mechanistic understanding of how autoregressive models represent and predict is a foundational scientific contribution to the field.
> > >
> > > ### Linear Extrapolation vs Regression
> > > We wish to clarify an important distinction between two types of linearity discussed in the paper. The temporal straightening hypothesis proposes that models extrapolate trajectories approximately linearly. The linear regression is simply our statistical tool for quantifying the association. A modest $r$ reflects that curvature is a scalar compression of a high-dimensional process, not a failure of the extrapolation hypothesis.
> > >
> > > ### Computational Neuroscience
> > > Further evidence that LLMs learn temporal straightening points to a convergence between biological and artificial systems. It suggests that transforming non-linear sequences into straightened trajectories is a normative computational principle for building predictive representations. Furthermore, our finding that trajectory curvature directly predicts model entropy provides a concrete geometric hypothesis for systems neuroscience: it suggests that biological networks might resolve predictive uncertainty by smoothing population manifolds. LLMs also offer a highly tractable testbed. By perturbing their trajectories, we can generate and test mechanistic hypotheses about analogous predictive coding and uncertainty resolution in the brain.
> > >
> > > **References**
> > >
> > > Hosseini, E., & Fedorenko, E. (2023). Large language models implicitly learn to straighten neural sentence trajectories to construct a predictive representation of natural language. Advances in Neural Information Processing Systems 36.

---

### Official Review · Reviewer_Fbv8 · 2026-03-12

**Soundness:** 3
**Presentation:** 3
**Significance:** 2
**Originality:** 3
**Overall Recommendation:** 6
**Confidence:** 4

**Summary:**

This paper investigates the relationship between a geometric property of transformer hidden-state trajectories — contextual curvature — and next-token prediction entropy. Building on prior work showing that LLMs progressively straighten representational trajectories across layers, the authors provide a direct, causal link between this geometry and model behavior. The key contributions are: (1) a cross-validated regression showing that middle-layer curvature predicts token-level entropy; (2) a training dynamics analysis showing this coupling emerges over the course of pretraining; (3) perturbation experiments establishing causal specificity; and (4) curvature regularization during training that modestly reduces output entropy without degrading validation loss.

**Compliance With Llm Reviewing Policy:**

Affirmed.

**Final Justification:**

The rebuttal addressed my main concerns and I think that this is a much more interesting approach than engaging in the typical bakeoff of minor improvements.

**Key Questions For Authors:**

Were any experiments conducted with test time straightening or consideration of output decoding guided by curvature?
were any experiments conducted where you placed curvature penalties on other layers to evaluate impact of tangling/untangling on less correlated layers?

**Limitations:**

Yes.

**Strengths And Weaknesses:**

Strengths
Originality and Significance. This paper stands apart from the large volume of capability-benchmarking work that dominates current LLM research. Geometric interpretations of learned representations are an important and underexplored direction, and this paper makes a meaningful contribution to that space. Rather than advancing the "bakeoff" of model comparisons, it offers genuine mechanistic insight into what internal representations are doing functionally. The connection drawn between temporal straightening — a principle borrowed from computational neuroscience — and the output uncertainty of autoregressive language models is creative, well-motivated, and well-executed.
The work raises an especially tantalizing practical question: is this a low- or no-cost method by which a deployed model could be made more certain of its responses? The perturbation experiments hint that middle-layer curvature could potentially be reduced post-hoc, without retraining, to selectively reduce uncertainty that arises from tangled representations rather than genuine linguistic ambiguity. This possibility is not fully explored, but it is a compelling direction that the community will want to follow up on.

Soundness. The methodological choices are generally well-motivated. Using the angle between adjacent difference vectors to capture trajectory bending is geometrically sensible: the difference vector from token A to token B and from B to C, with the angle between them quantifying the bend at B, directly captures local trajectory curvature in a coordinate-free manner. The cross-validated regression framework is appropriate. The use of matched perturbation magnitudes across intervention types (Table A1) to isolate geometric orientation from perturbation scale is a particularly careful design choice and substantially strengthens the causal claims. The training dynamics analysis using Pythia checkpoints is well-suited to the question and provides a compelling developmental account.

Presentation. The overall narrative arc is clear and easy to follow. Figure 1 is an effective schematic that orients the reader before the technical sections. The progression from correlation to training dynamics to causal perturbation to regularization is logical and well-paced.

Weaknesses and Requested Revisions
The geometric formalism needs more exposition. The definition of the difference vector and the angle-based curvature measure will not be immediately transparent to all readers. It should be made explicit why the angle between adjacent difference vectors is the right quantity — specifically, that vₖ = xₖ₊₁ − xₖ represents the displacement from token k to token k+1 in activation space, and that the angle between vₖ and vₖ₊₁ captures the bend induced at position k+1. A reader unfamiliar with discrete curve geometry may not reconstruct this from the notation alone. A one- or two-sentence geometric gloss, perhaps alongside Figure 1C, would help considerably.

Window size sensitivity should be more thoroughly reported. The paper justifies a backward-looking window of size three by noting that larger windows do not add predictive power. However, results for other window sizes (at minimum w=1, 2, 5) should be reported, either in the main text or in the appendix. It is important to know not just whether larger windows fail to improve performance, but whether performance degrades, and over what range the measure is stable. This bears directly on the robustness of the curvature definition.

Effect sizes are modest and this deserves more discussion. The peak Pearson correlation of approximately r ≈ 0.15 is statistically reliable and consistent across model families, but the paper's framing somewhat understates how much variance in entropy is left unexplained. The authors briefly acknowledge this, attributing it to curvature being a scalar summary of a high-dimensional representation. This is reasonable, but the discussion would benefit from a clearer treatment of what additional geometric features — intrinsic dimensionality, path length, local subspace orientation — might complement curvature and push prediction further.

Regularization effects are small. The curvature regularization experiments show the expected directional effects but with modest magnitude. The authors attribute this to model and data scale, which is plausible. It would strengthen the paper to discuss what scale of model or training data would be needed to expect larger effects, or whether the regularization weight schedule might be better optimized.

Minor Comments
The distinction between the long-context (LAMBADA) and short-context (Universal Dependencies) datasets is well-motivated in principle, but the paper could more clearly report whether the curvature-entropy relationship differs meaningfully in magnitude between the two datasets, and offer an interpretation of any differences.

The claim that validation loss is similar across the baseline, untangled, and tangled conditions is important for interpreting the regularization results. A formal statistical comparison, rather than visual inspection of Figure 5C inset, would make this more convincing.

---

> ### Author Rebuttal · Authors · 2026-03-31
>
> We thank the reviewer for the insightful comments and included additional analysis based on their suggestions.
>
> ### Error in paper:
> *[Figure 4 Corrected Analysis](https://drive.google.com/file/d/1ST3Do4Te74VSmiEaFkV7bCbKvBEqPu5V/view?usp=sharing)*
>
> After submission, we found an error: Figure 4 reported perturbation results at layer 16 of Pythia-2.8B, but plotted layer 11. We ran the layer 16 analysis, which largely replicates layer 11: trajectory-subspace perturbations strongly couple ΔC and ΔH, while full-space and random-subspace effects are negligible. However, the planar perturbation's effect is weaker at layer 16. The specific subspace linking trajectory to extrapolation may vary across layers. We plan to investigate this, but our key observation that entropy is selectively sensitive to curvature changes along the trajectory subspace holds at both layers.
>
> Perturbing intermediate layers is motivated by more than just the strength of curvature-entropy coupling. At later layers, perturbations tend to shift not only the certainty of the output distribution but the location of the token distribution (measured via rank order correlation between baseline and perturbed model outputs). At intermediate layers, perturbations more cleanly isolate changes in entropy: they flatten or sharpen the distribution without reordering it.
>
> ### Geometric Formulas:
> We agree further exposition on the geometric formalism would help. We will add the following to Section 2.3 alongside Figure 1C: "Each difference vector vₖ = xₖ₊₁ − xₖ represents the displacement from token k to token k+1 in activation space. The angle between adjacent difference vectors vₖ and vₖ₊₁ then captures how sharply the trajectory changes direction at position k+1: small angles indicate the trajectory continues roughly straight, while large angles indicate a sharp bend.”
>
> ### Effect sizes:
> Temporal straightening implies that the model is shaping the geometry of the representational manifold to make trajectories easier to extrapolate from. Our curvature measure captures one aspect of this, but predictability on a manifold is a richer problem: it also depends on whether nearby trajectories are converging toward similar predictions or diverging toward different ones, how many dimensions the trajectory occupies, and how quickly the representation moves through space. Richer manifold measures such as torsion, intrinsic dimensionality, path length, and Riemannian curvature could explain variance that a single scalar misses. The consistency of the effect across models and datasets, and the stronger per-token effects revealed by the perturbation experiments, suggest the underlying geometric relationship is richer than the scalar correlation alone indicates.
>
> ### Small regularization effects:
> We explored a range of regularization multiplier values and annealing schedules. The reported value was approximately the strongest that did not degrade validation loss. Higher values destabilized training. This suggests the modest effect size partly reflects a tradeoff between geometrical simplicity and output predictivity: curvature regularization can only push geometry so far before collapsing the representation. Larger models may have more representational capacity to accommodate geometric constraints without sacrificing performance, but we cannot make strong claims about what scale is needed. We think this is worth pursuing given the consistent directionality of our results. Indeed recent work by Huang, et al. 2026 has shown promising results during post-training with similar regularizers.
>
> ### Regularizing other layers:
> We did conduct experiments regularizing layers beyond 7 and 8. In most cases it was not possible to reduce curvature at these layers without degrading validation loss, which is itself informative. It suggests that the middle layers where curvature-entropy coupling is strongest are also where representational geometry is most amenable to being reshaped without disrupting the model's predictive capacity.
>
> ### Test time straightening:
> Hosseini & Fedorenko (2023) included preliminary analyses showing that mean curvature of sentences produced by a model is lower than natural sentences, but this was observational rather than an active decoding intervention. The idea of using curvature as a signal to guide output decoding at test time, and of evaluating changes in curvature as text is outputted during chain of thought, are compelling directions that follow directly from our results, and ones we plan to pursue.
>
> ### Window Sizes:
> *[Figure: contextual curvature window Sizes](https://drive.google.com/file/d/1ncQIjsVpSujlXzulkv-P8cV1PcSOeRPw/view?usp=sharing)*
>
> We built and cross validated regressions from contextual curvature to entropy for window sizes w=1,2,5,10 to support our use of w=3.
>
> ### References:
> * Huang, H., LeCun, Y., & Balestriero, R. (2026). Semantic Tube Prediction: Beating LLM Data Efficiency with JEPA. arXiv preprint arXiv:2602.22617.

---

> > ### Author Rebuttal · Reviewer_Fbv8 · 2026-04-04
> >
> > I did and do like this paper. Everything was thoughtfully addressed and review was raised.

---

> > > ### Author Response · Authors · 2026-04-08
> > >
> > > We are very grateful for the positive feedback and the confirmation that all concerns were fully resolved. We’d like to thank the reviewer for their time and interest in our work.

---

### Decision · Program_Chairs · 2026-04-30

**Decision:**

Accept (regular)

**Comment:**

This paper studies how the geometry of hidden-state trajectories in autoregressive LLMs relates to model uncertainty. In particular, it shows that contextual curvature of residual-stream trajectories predicts next-token entropy, that this coupling emerges over training, that trajectory-aligned perturbations causally modulate entropy, and that curvature regularization can modestly reduce uncertainty without harming validation loss.

I recommend acceptance. The paper offers a clear and interesting mechanistic contribution by linking an interpretable geometric property of representations to token-level behavior, and it does so with a well-structured empirical package: cross-model analysis, training-time dynamics, causal perturbations, and training interventions. The reviewers found the work technically solid and meaningful, and the main concerns raised in review were addressed in rebuttal, including additional analyses on window size, longer contexts, geometric interpretation, and the corrected perturbation analysis.

While the effect sizes are modest and the claims should remain appropriately scoped, the evidence is consistent across datasets, model families, and interventions, and the paper advances an underexplored but important direction for understanding prediction in language models. Overall, this is a strong ICML paper with genuine insight and good follow-through in the rebuttal, and I recommend acceptance.